# Early Strokes Are Associated with More Global Cognitive Deficits in Adults with Sickle Cell Disease

**DOI:** 10.3390/jcm12041615

**Published:** 2023-02-17

**Authors:** Maryline Couette, Stéphanie Forté, Damien Oudin Doglioni, Armand Mekontso-Dessap, David Calvet, Kevin H. M. Kuo, Pablo Bartolucci

**Affiliations:** 1Sickle Cell Referral Centre–UMGGR, University of Paris Est Créteil, Henri Mondor APHP, 94010 Créteil, France; 2CARMAS (Cardiovascular and Respiratory Manifestations of Acute Lung Injury and Sepsis), University of Paris Est Créteil, 94010 Créteil, France; 3IMRB, INSERM, University of Paris Est Créteil, 94010 Créteil, France; 4Department of Medecine, Centre Hospitalier de l’Université de Montréal, Montréal, QC H2X 0C1, Canada; 5Department of Medicine, Université de Montréal, Montréal, QC H3C 3J7, Canada; 6Centre de Recherche du CHUM, Montréal, QC H2X 0A9, Canada; 7Laboratoire Inter-Universitaire de Psychologie—Personnalité, Cognition, Changement Social (LIP/PC2S), Université Grenoble Alpes, 38058 Saint-Martin-d′Hères, France; 8INSERM, UMR 1266, Psychiatry and Neurosciences Institute of Paris, Paris-Descartes University, Department of Neurology and Stroke Unit, Sainte-Anne Hospital, 75014 Paris, France; 9Department of Medicine, University Health Network, Toronto, ON M5G 2C4, Canada; 10Department of Medicine, University of Toronto, Toronto, ON M5S 1A8, Canada; 11INSERM-U955, Equipe 2, Laboratoire d’Excellence, GRex, Institut Mondor, 94000 Créteil, France

**Keywords:** sickle cell disease, sickle cell anemia, stroke, neuropsychological assessment, cognitive impairment, cognitive deficits

## Abstract

This study sought to link neurocognitive profiles in sickle cell disease (SCD) patients with clinical characteristics. We conducted a prospective cohort study of adults with SCD who underwent comprehensive neuropsychological assessment at the UMGGR clinic at Henri Mondor Hospital, Créteil (France). A cluster analysis was performed based on neuropsychological testing scores. The association between clusters and clinical profiles was assessed. Between 2017 and 2021, 79 patients with a mean age of 36 [range 19–65] years were included. On principal component analysis, a 5-factor model presented the best fit (Bartlett’s sphericity test [χ^2^ (171) = 1345; *p* < 0.001]), explaining 72% of the variance. The factors represent distinct cognitive domains and anatomical regions. On hierarchical classification, three clusters emerged. Cluster 1 (*n* = 24) presented deficits in all five factors compared to Cluster 3 (*n* = 33). Cluster 2 (*n* = 22) had deficits in all factors, but to a lesser extent than Cluster 1. MoCA scores mirrored the severity of these cognitive deficits. Age, genotype and stroke prevalence did not differ significantly between clusters. However, the time of first stroke occurrence differed significantly between Cluster 1 and 2–3: 78% of strokes occurred during childhood, whereas 80% and 83% occurred during adulthood in Clusters 2 and 3, respectively. Educational attainment was also reduced in Cluster 1. SCD patients with childhood stroke seem to be at increased risk of a global cognitive deficit profile. In addition to existing methods of primary and secondary stroke prevention, early neurorehabilitation should be prioritized in order to reduce the long-term cognitive morbidity of SCD.

## 1. Introduction

Sickle cell disease (SCD) is the most common inherited blood disorder in France [1]. On a global scale, the prevalence has increased gradually, as has life expectancy, with advancements in care over the last decades [1,2,3]. As the number of adult patients with SCD is growing, management of chronic organ complications is emerging as a challenging new field of clinical research. Stroke is one of the most devastating complications of SCD, and it occurred in as many as 24% of patients by age 45 [4]. Silent cerebral infarcts (SCI) are found in 39% of patients by age 18 and in 53% of adult patients [5,6,7]. In adults, the estimated prevalence of cognitive impairment is 33% [8]. These cognitive deficits impact quality of life and social functioning, and they often lead to unemployment [9].

There are many studies on the cognitive functioning of children with SCD, since the major issue is schooling and the proper conduct of education. In adulthood, the interest in cognition seems to decrease with sparse published studies as we can notice in the meta-analysis of Prussien et al. on cognitive functioning in SCD patients (only 7 studies on SCD adult patients against 103 studies on children) [10]. This lack was also observed in Sahu’s review, where the majority of studies are on children or young adults [11]. However, this issue is also important, as the SCD population is getting older, and cognitive disorders make medical follow-up and social support more difficult. In addition, patients are faced with personal and medical situations that require them to make important decisions with far-reaching consequences.

In Prussien meta-analysis, authors reported lower processing speed and executive functioning deficits in both children and adults with SCD [10]. In a recent comprehensive review, Sahu et al. reported mainly dysexecutive disorders with learning difficulties, working memory disorders and attentional disorders in adults and children, with dramatic consequences both in their daily and professional lives, but also in terms of therapeutic follow-up. Cognitive disorders are more important in patients with cerebro-vascular disorders like stroke and/or SCI or vasculopathy [11,12]. However, cognitive disorders were depicted in adult patients with SCD, even in the absence of overt neurological events, in a neuroimaging study [13]. Indeed, SCD patients exhibited a thinner frontal lobe and a reduced basal ganglia and thalamus with a marked reduction of processing speed and, to a lesser extent, deficiencies in working memory and perceptual organization. In accordance with frontal/subcortical damages or dysfunctions, a recent cross-sectional study on 22 adult SCD patients reported dysexecutive deficits [14].

The American Society of Hematology now recommends cognitive surveillance in children and adults with SCD [15]. However, the optimal surveillance strategies have not been established. Since access to comprehensive neurocognitive testing is limited, short screening tools can help identify patients at risk of cognitive impairment. For instance, the Montreal Cognitive Assessment (MoCA) and Rowland Universal Dementia Assessment Scale (RUDAS) have been proposed for cognitive screening in adults with SCD [16,17,18].

We hypothesized that the cognitive profile of adults with SCD varies with the presence of prior neurological events (stroke and/or SCI) and that the MoCA can be used as a screening tool for the identification of these different cognitive profiles. The first objective of this study was to derive different cognitive profiles from the complete neuropsychological data of adult patients with SCD. The second objective was to differentiate these patient groups based on demographics, medical history, and MoCA performance. The third objective was to highlight the possibility of rapidly screening patients with cognitive disorders through a rapid comprehensive test such as the MoCA. 

## 2. Methods

### 2.1. Participants

We conducted a prospective study on 79 patients with SCD recruited at a single center (the referral center UMGGR at Henri Mondor Hospital, France). Patients were enrolled locally from the PCDREP (Perfusion Cébébral DREPanocytose) cohort. In summary, PCDREP is a French prospective, multicenter, observational cohort study that aims at characterizing adults with SCD at high risk of cerebrovascular disease. Since 2011, adults with SCD were included if they had a previous stroke, transient ischemic attack, seizure and/or presented definite or probable intracranial vasculopathy. This cohort received approval from the ethics committee (Comité de Protectiondes Personnes Île-de-France Saint-Louis). Demographics, clinical history, laboratory findings and medical treatments were recorded prospectively at inclusion in a database.

In order to analyze a diverse group of patients with and without cerebrovascular disease, we also recruited adults with SCD at the UMGGR. Participants were adults of any SCD genotype who were able to provide informed consent and follow study instructions.

All of the participants underwent standard comprehensive neuropsychological examination and a global cognitive functioning assessment, and SCD patients received a psychological examination during their routine care follow-up.

### 2.2. Clinical and Neuropsychological Assessments

The comprehensive neuropsychological assessment included different tests assessing the cognitive domains proposed by the VasCog statement for vascular cognitive disorder [19] (see Table 1). The cognitive battery combined standardized neuropsychological pen–paper tests that have an established utility in clinical practice. Memory skills were assessed by the free and cued recall test Grober and Buschke for verbal episodic memory and by the recall of a Rey–Osterrieth complex figure (ROCF) for visual memory. Working memory was assessed by the Number Memory Subtest of the WAIS IV (Wechsler Adult Intelligence Scale). The Symbol Digit Modality Test (SDMT) version A was used to assess processing speed. Attentional processes (selective attention and sustained attention) were measured with the attentional D2 test. Executive functioning was assessed by three tests: Trail Making Test (TMT) A and B, Stroop Test and verbal fluencies (semantic and phonemic: animal and letter P in two minutes) [20,21,22,23,24,25,26,27,28]. The Rey–Osterrieth Complex copy was used to evaluate planning and visuo-constructional praxis. Last, language was assessed by an oral naming French test (Dénomination orale 40 items: DO40) from a semantic assessment battery BECS-GRECO [29].

The MoCA test was used to screen for cognitive impairment [20]. Depression and anxiety levels known to disturb cognition were measured with the Hospital Anxiety and Depression Scale (HADS) [30]. Indeed, in a recent meta-analysis, Semkovska et al. identified 252 studies assessing cognition during a major depressive episode and showed a frontal/subcortical profile (selective attention, working memory and learning) depicted by SCD patients [31].

### 2.3. Outcome Definition

MoCA scores were defined according to the MoCA manual for scoring. Any score less than 26 was defined as suggestive of neurocognitive damages [20].

In terms of neuropsychological measures, scores without standardized transformation were included in analysis.

On HADS screening, a score between 8 and 10 was suggestive of anxiety or depression. Scores above 10 were indicative of anxiety or depression [30].

### 2.4. Variables

The clinical and demographic variables were age, sex, SCD genotype, geographical origin, history of stroke, timing of first stroke and presence of vasculopathy and/or white matter hyperintensities. Stroke was confirmed by medical history and imaging. Timing of first stroke was classified as occurring in adulthood (≥18 years of age) or childhood (<18 years of age). White matter hyperintensities were noted in imaging reports. Cerebral vasculopathy was defined by cerebral MRI and transcranial doppler. The highest level of education (HLE) was scored based on the number of years of education post kindergarten. Geographical origin was self-reported and categorized as France/Overseas France, Africa or India.

### 2.5. Statistical Tests

Principal component analysis (PCA) was used to gather variables in meaningful factors. Bartlett’s Test of Sphericity was used to test that factors were not independent. The quality of inter-item correlations was tested using the Kaiser–Meyer–Olkin approach. Items loading greater than 0.4 were assigned to a specific factor. Split items were assigned to a specific factor if the square of the loading for a factor was greater than 50% of its loading on any other factor. When the procedure could not lead to factor assignment, the split items’ contribution to the reliability of each possible subscale was examined. Based on these factors, an unsupervised hierarchical clustering analysis of principal components (HCPC) was performed in order to group individuals into meaningful classes of cognitive function.

Multiple imputation was performed using factorial analysis derived from principal component analysis. The MissMDA library on R was used [32].

PCA and hierarchical classification of the principal component were conducted using R software [33]. ANOVA (unidirectional Kruskall–Wallis) and post-hoc Dwass, Steel, Critchlow and Fligner peer-to-peer comparisons were made using Jamovi.

### 2.6. Ethical Issues and Approval

Written informed consent was obtained from all of the participating patients. This study was approved by the local ethics committee (Protocol 2013/NICN).

## 3. Results

### 3.1. Participant’s Characteristics

We included 79 individual patients whose characteristics are detailed in below in Table 2.

### 3.2. Principal Component Analysis

Of the 79 patients, 66 (84%) had completed all tasks of the neurocognitive assessment. Principal component analysis of the neurocognitive test results with varimax rotation identified five distinct factors (see Appendix A for further information on the initial eigenvalues and scree plot). The total communality of the model is large (h^2^ = 13.68), indicating that the variance shared by all variables is substantial and allowing for a factor analysis with a small sample size [34]. Bartlett’s Test of Sphericity indicates that the factors are not independent [χ^2^ (171) = 1345; *p* < 0.001], meaning that they all refer to the same construct, namely neuropsychological functioning. The quality of inter-item correlations was considered to be good (Kaiser–Meyer–Olkin = 0.77), meaning that each test was associated with its factor. The final factorial structure proposal explained 72% of the variance of the neuropsychological functioning.

The first factor encompassed tests of visual attention and visual organization (referring to the right hemisphere). The second factor seemed to depict the mental/cognitive control (frontal lobe functioning), the third was ascribed to language and memory (left temporal lobe functions), the fourth factor encompassed tests of selective inhibition or selective attention (fronto-parietal component) and the last referred to shifting skill (subcortical loop).

### 3.3. Hierarchical Classification on Principal Component

Each patient was projected into a five-dimensional space, according to the five factors. Three clusters emerged from this modelling, with 24 patients for Cluster 1, 22 patients for Cluster 2 and 33 patients for Cluster 3. Neuropsychological assessments between clusters are reported in Table 3.

Results of all the different neurocognitive tasks differed significantly between clusters, as confirmed by ANOVA, except for the TTMT B-A errors task. Overall, Cluster 3 had the best performances on the different tasks. As a result, Cluster 3 was used as the reference group.

Post-hoc analysis considering Cluster 3 as normal revealed that Cluster 1 was deficient in all factors. Deficits were visible in processing speed (GZ index of D2 and SDMT score), sustained attention (GZ-F index of D2), visual attention and organization (TMTA time and ROCF copy), elective and focalized attention (Stroop inhibition time, F% and KL indices of D2), working memory (WAIS IV reverse span and number memory), inhibition abilities (errors in TMTA), concept generation (phonological fluency), language (semantic fluency and naming), verbal memory (Grober and Buschke total recall) and shifting (time in TMT B-A). Compared to Cluster 3, Cluster 2 also demonstrated reduced performance, but to a lesser extent. Additionally, not all tasks showed reduced performance of Cluster 2 as compared to Cluster 3; in particular, verbal memory and visual organization performance were similar for Clusters 2 and 3 (Table 3).

### 3.4. Characterization of Clusters

Comparisons between clusters regarding clinical MoCA scores and characteristics are shown in Table 4 and Table 5.

#### 3.4.1. MoCA

On the MoCA test, Cluster 1 showed a lower mean score relative to Cluster 2 (*p* < 0.001) and Cluster 3 (*p* < 0.001). A mean score of 20 (2) indicates probable neurocognitive disorders [20]. Cluster 2 had a mean score of 23 (3), indicating possible mild neurocognitive impairment. Finally, Cluster 3 had a mean score of 26 (2), suggesting no or few cognitive disorders (Table 4).

#### 3.4.2. Demographics and Baseline Characteristics

There was no significant age difference between clusters (*p* > 0.05). Additionally, measures of depression and anxiety levels did not differ significantly between clusters (*p* > 0.05). Cluster 1 had a lower educational level than the other two clusters (F(2.76) = 13.46; *p* < 0.001). Finally, there were more individuals of French origin in Cluster 3, while this proportion was the lowest in Cluster 2 (see Table 5).

The proportion of those with the HbSS genotype seemed higher in Cluster 1. However, this was not statistically significant.

#### 3.4.3. Cerebral Abnormalities

The Pearson Chi-square analyses of the three clusters were not significant for the presence of white matter abnormalities (*p* > 0.05), vasculopathy (*p* > 0.05) or stroke occurrence (*p* > 0.05) (see Table 5). Regarding the time course of stroke occurrence, in Cluster 1 strokes occurred during childhood in 78% of the patients, whereas 80% and 83% occurred during adulthood in Cluster 2 and Cluster 3, respectively (see Table 5).

## 4. Discussion

The main objective of this study was to identify cognitive profiles in reference to vascular cognitive dementia criteria (VASCOG), and our hypothesis was that this profile varies with the presence of overt neurological abnormalities (overt stroke or SCI). Our PCA on comprehensive cognitive battery finely assessed cognition in SCD adults. We identified five factors representing all cognitive domains: left hemisphere, right hemisphere, frontal lobe and subcortical loops functioning. We identified three clusters of SCD patients depicting three different profiles. Cluster 3 was considered normal or less affected, Cluster 2 depicted the classical cognitive profile (lower processing speed and deficits in frontal lobe functioning) and Cluster 1 was critically altered in all cognitive domains and showed the frontal/subcortical profile and some cortical alterations like memory, language and visual organization disorders [10]. These cognitive alterations raise the question of the possible inclusion of SCD in the spectrum of vascular cognitive dementia or at least in vascular cognitive disorders by their multiple vascular mechanisms: large and small vessel lesions, ischemic or hemorrhagic stroke and cerebral hypoperfusion with cerebral vasculopathy [35,36,37].

The MoCA test could be considered a reliable screening tool in routine care, as the score and the subtest’s damages quickly indicated the future cognitive profile. A recent review on fifteen studies from Ghafar and collaborators indicated that the MoCA test had excellent accuracy in differentiating vascular dementia from controls in addition to good internal consistency. They also showed that MoCA had great accuracy when separating vascular mild cognitive impairment [38]. Given our present results, we also consider it an excellent tool in routine care to quickly detect adult SCD patients with mild cognitive impairment and which profile they fall into (frontal/subcortical and/or cortical damages). Indeed, a patient with a total adjusted score greater than or equal to 26 could be considered cognitively unimpaired, a patient with a total adjusted score between 26 and 20 could be classified as mildly impaired, and a patient with a score less than 20 could be considered severely impaired with possible cortical damage.

Our results showed that adults with the most severe neurocognitive deficits had more often experienced stroke during their childhood (78%) and achieved lower educational levels. Childhood strokes may have disrupted the schooling of these patients, explaining the lower educational attainment. In children, the presence of cerebral vasculopathy is the leading cause of ischemic stroke, while the causes are more diverse in adults [39]. Chronic transfusions are considered the standard treatment in children with vasculopathy in order to decrease the risk of a first ischemic stroke [40,41]. Early tailored neurorehabilitation programs should be easily accessible to children with SCD suffering from ischemic stroke and severe vasculopathy as a means to mitigate the impact on cognition and social functioning. Unfortunately, the socioeconomic status of the family was one of the major factors impacting full IQ in a recent meta-analysis conducted by Prussien and collaborators, and it lead to the disruption of schooling [42].

The limitations of our study include missing data on patients who missed their cognitive assessment. Among patients who had their assessment, some were not able to perform all of the tests. The principal reason was fatigability and bad visual accuracy. Cluster 3, depicted as cognitively spared, has a higher level of education, but it should be noted that the majority of patients in this cluster are from France, and access to education and accommodation is often easier in France than in other countries, which may explain the higher level of education. However, this also aligns with our suggestions, namely that the disease disrupts the course of studies, and that this is necessarily worse in countries which do not have the possibility of setting up specific arrangements in addition to appropriate rehabilitations. Another limitation is that we included all SCD genotypes, but we know that stroke mechanisms vary between them: hemorrhagic strokes are frequent in adults with SS genotypes, while both ischemic and hemorrhagic strokes are frequent in adults with SC genotypes. SC adults are in majority older, a fact which could also be a limitation, but age was not significantly different between clusters, and SS genotypes were similarly distributed between clusters. In addition, cultural bias is well known within the SCD population, and there is a specific need to use adapted cognitive tools [16]. Finally, fatigue and pain have repercussions on cognitive functioning, as previously demonstrated in fibromyalgia [43]. Chronic fatigue also has a considerable impact on the cognition of young people [44]. Fatigue and pain can aggravate cognitive disorders, culminating in depression and anxiety [45]. Unfortunately, our neuropsychological battery lasted around two hours, a duration that is probably too long. A shorter cognitive battery would be necessary for SCD patients often experiencing pain and chronic fatigue syndrome. Finally, our Cluster 3, which was considered cognitively normal, had more people of French origins than Cluster 2 and Cluster 1. Whereas the comprehensive battery is feasible in patients from Africa, we must be attentive to African patients with childhood stroke histories. Adapted cognitive tests which limit cultural and educational bias are needed.

## 5. Conclusions

This study describes for the first time three different cognitive profiles in adults with SCD: one without major cognitive deficits, another with the classical frontal/subcortical profile previously described, and one with global and more severe cognitive deficits. MoCA scores mirrored these profiles, supporting its role as a screening tool for cognitive impairment in SCD. Childhood strokes were associated with the more global and severe cognitive deficits and the lowest educational achievement. This study highlights the importance of stroke prevention in childhood and the provision of early cognitive support to help these patients succeed in school.

## Figures and Tables

**Table 1 jcm-12-01615-t001:** Cognitive battery and VASCOG domains.

VASCOG Domains	Cognitive Tests
Attention and processing speed	D2 SDMT
Frontal-Executive Function	NM from WAIS-IV TMT A & B Stroop ROCF (copy) Verbal fluencies
Learning and Memory	Grober & Buschke test ROCF (memory
Language	DO40 from BECS Verbal fluencies
Visuo-Constructional-Perceptual Ability	ROCF TMT SDMT D2
Praxis-Gnosis-Body Schema	DO40 ROCF (copy)

Abbreviations: D2 = D2 Test of Attention, DO40: Dénomination orale 40 items: NA = not assessed, NM = Number Memory Subtest, ROCF = Rey–Osterrieth Complex figure, SD = standard deviation, SDMT = Symbol Digit Modalities Test, TMT = Trail Making Test, VASCOG = the International Society of Vascular Behavioral and Cognitive Disorders, WAIS-IV = Wechsler Adult Intelligence Scale version IV.

**Table 2 jcm-12-01615-t002:** Characteristics of the patients included in the prospective analysis.

	Patient Characteristics *n* = 79
Age in years, mean (range)	36 (19–65)
Male, *n* (%)	39 (49%)
Highest level of education in years, mean (range)	12 (5–17)
SCD genotype, *n* (%)	
HbSS	62 (78%)
HbSC	12 (15%)
HbSβ0	3 (4%)
HbSβ+	1 (1%)
Other sickling genotype *	2 (2%)
History of stroke, *n* (%)	28 (35%)
Cerebral vasculopathy, *n* (%)	33 (42%)
White matter hyperintensities, *n* (%)	26 (33%)
Geographical origin, *n* (%)	
France/Overseas France	36 (46%)
Africa	41 (52%)
India	1 (1%)

* Other sickling genotypes include HbSO-Tibesti and one individual with missing information about the precise genotype. Abbreviations: SCD = sickle cell disease. Missing data: cerebral vasculopathy 1%, geographical origin 1%, history of stroke 1% White matter hyperintensities, 1%.

**Table 3 jcm-12-01615-t003:** Clusters’ performance on individual neurocognitive tasks. Individual tasks are organized according to the five factors identified from the principal component analysis. For each cluster, the mean scores (with standard deviation) on individual tests are presented. Cluster 3 was defined as the reference group for inter-cluster comparisons.

	Cluster 1 (*n* = 24)	Cluster 2 (*n* = 22)	Cluster 3 (*n* = 33)	ANOVA ^&^ *p* Value
**Factor 1: visual attention and organization** (**right hemisphere**)				
D2 test GZ index (processing speed)	292.50 *** (76.26)	423.62 (72.57)	469.94 (76.41)	<0.001
D2 test GZ-F index (sustained attention)	265.29 *** (71.97)	398.09 * (57.57)	451.03 (72.53)	<0.001
ROCF memory (visual memory)	16 * (7)	14.93 ** (6.26)	20.47 (5.97)	0.005
SDMT score (processing speed)	21.45 *** (6.25)	30.41 *** (6.79)	45.15 (8.67)	<0.001
TMT A time in sec (visual research)	84.54 *** (30.05)	55.77 *** (14.67)	35.79 (10.28)	<0.001
ROCF copy (visual organization)	25.92 *** (7.29)	28.68 (8.35)	33.21 (2.15)	<0.001
**Factor 2: mental control** (**frontal lobe**)				
WAIS IV Number memory score (working memory)	3.50 *** (1.67)	5.50 ** (1.34)	7.42 (2.08)	<0.001
WAIS IV Reverse span (working memory)	2.54 *** (0.83)	3.14 * (0.64)	3.88 (1.08)	<0.001
Phonemic fluency (concept generation)	9.37 *** (5.14)	17.36 * (7.09)	24.73 (8.77)	<0.001
Stroop Inhibition errors (inhibition)	3.66* (4.97)	2.27 (4.96)	0.66 (1.33)	0.026
Stroop Inhibition time in sec (focalized attention)	113.87 *** (52.52)	86.36 *** (28.80)	53.66 (20.60)	<0.001
D2 test F% score (selective attention)	9.86 ** (6.71)	5.62 (5.21)	4.12 (2.14)	0.003
**Factor 3: Language and memory** (**left hemisphere**)				
Semantic fluency (semantic)	16.54 *** (5.48)	21.54 *** (5.73)	30.18 (7.34)	<0.001
DO40 naming (semantic)	28.92 *** (7.29)	32.41 ** (6.57)	37.27 (2.56)	<0.001
Grober & Buschke total recall (verbal memory)	38.74 *** (9.09)	45.04 (4.29)	46.12 (4)	<0.001
**Factor 4: Focalized attention** (**fronto-parietal lobe**)				
TMT A errors (inhibition)	0.79 ** (1.21)	0.23 (0.43)	0.09 (0.29)	0.003
D2 test KL index (selective attention)	75.45 *** (40.20)	127.86 ** (41.72)	162.72 (33.15)	<0.001
**Factor 5: Reactive mental flexibility** (**subcortical loop**)				
TMT B-A time in sec (shifting)	182.54 *** (92.48)	95.50 *** (46.96)	57.91 (40.51)	<0.001
TMT B-A errors (shifting)	2.00 (2.19)	0.77 (0.97)	1.18 (1.51)	0.185

^&^ Unidirectional Kruskall–Wallis ANOVA was performed. For post-hoc analysis, Cluster 3 was the reference group and comparisons were made between Cluster 1 and 3 and between Cluster 2 and 3. * denotes a *p*-value < 0.05; **denotes a *p*-value < 0.01; *** denotes a *p*-value < 0.001.

**Table 4 jcm-12-01615-t004:** Clusterts’ MoCA scores. Adjustments were made for educational levels ≤ 12 years according to the MoCA scoring manual.

	Cluster 1 (*n* = 24)	Cluster 2 (*n* = 22)	Cluster 3 (*n* = 33)	ANOVA *p* Value	Post-Hoc Analysis ^&^		
					1 vs. 2	1 vs. 3	2 vs. 3
Total score, mean (SD)	18.75 (2.51)	22.32 (2.92)	25.51 (2.29)	<0.001	<0.001	<0.001	<0.001
Total score adjusted, mean (SD)	19.58 (2.48)	22.77 (2.94)	25.88 (2.23)	<0.001	<0.001	<0.001	<0.001

^&^ Bonferroni correction. Abbreviations: SD = standard deviation.

**Table 5 jcm-12-01615-t005:** Characterization of clusters in terms of different variables using Pearson Chi-square, ANOVA and post-hoc analyses.

	*n*	Cluster 1 *n* = 24	Cluster 2 *n* = 22	Cluster 3 *n* = 33	*p* Value
Mean age (SD)	79	35.75 (10.44)	39.12 (15.07)	34.40 (9.14)	0.33
Ethnic origin (France)	79	41.67%	22.73%	63.64%	0.023
Mean Educational Level (SD)	79	10.04 *** (2.93)	12.68 (2.83)	13.57 (2.09)	>0.001
HbSS genotype	79	87.50%	72.73%	72.73%	0.659
White Matter Abnormalities	79	29.17%	27.27%	39.39%	0.578
History of stroke	79	41.67%	27.27%	36.36%	0.589
Childhood stroke	26	77.78% *	20.00%	16.67%	0.011
Cerebral vasculopathy	79	54.17%	36.36%	36.36%	0.34
Mean HADS Anx (SD)	72	6.59 (3.18)	8.72 (3.87)	6.76 (4.15)	0.07
Mean HADS Dep (SD)	72	4.23 (3.15)	7.00 (3.09)	5.61 (4.17)	0.13

Post-hoc analysis of Cluster 1 versus 3 and Cluster 2 versus 3: * denotes a *p*-value < 0.05; *** denotes a *p*-value < 0.001. Abbreviations: HADS = Hospital Anxiety and Depression Scale, SD = standard deviation.

## Data Availability

The data presented in this study are available on request from the corresponding author. The data are not publicly available due to privacy restrictions.

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
