# Peer review of "Early Strokes Are Associated with More Global Cognitive Deficits in Adults with Sickle Cell Disease"

_jcm, 2023, doi:10.3390/jcm12041615_

Round 1

Reviewer 1 Report

The manuscript is interesting and well-written. Some suggestions were made for improving the article:

 1. Add references in lines 55-61 to confirm those affirmations.

2. Replace  "Montreal Neurocognitive Assessment (MoCA)" with "Montreal Cognitive Assessment (MoCA)" in line 75.

3. Table 1: add the DO40 to the abbreviations list.

4. Add references to the outcome definition (lines 132-138).

5. Review the citation of the statistical test (lines 161-163) in accordance with journal norms.

5. It is not necessary to repeat results in the table and text. Remove the values in the text (172-175).

6. Tables 3 and 5: it is missing two numbers after the commas.

7. The results of MoCA could be described in Table 2 and the post-hoc description could be in the text. Table 4 is not necessary.

8. The aims of the study should be reviewed (lines 78-84). One of the aims was to compare the cognitive performance of three clusters and the second aim was to compare variables (neurological and demographic) that differentiate the groups. 

9. You can not use the same MoCA cutoff for patients with lower and high schooling (lines 133-134). This can influence the results. Review the cutoff by schooling. 

10. The discussion is focusing on the results of the MoCA, but this is not informed previously in the aim, so this should be reviewed.

Author Response

We thank the reviewer for the constructive comments made and hope that the version we are submitting today will be satisfactory. We strongly believe that the review process has greatly improved the quality of our article. You will find a point-by-point answer to all the questions that have been raised.

  1. Add references in lines 55-61 to confirm those affirmations.

R1 : thank you for your remark; We added the reference of Prussien et al. (2019) already cited in the manuscript, which highlights the imbalance in the literature on cognition between children and adults. We add the reference as follows in the manuscript : “In adulthood, the interest in cognition seems to decrease with sparse published studies as we can notice in the meta-analysis of Prussien et al. on cognitive functioning in SCD patients (only 7 studies on SCD adult patients) [10].”

  1. Replace  "Montreal Neurocognitive Assessment (MoCA)" with "Montreal Cognitive Assessment (MoCA)" in line 75.

R2 : thank you for your notice, we apologize and we corrected it in the manuscript.

  1. Table 1: add the DO40 to the abbreviations list.

R3 : we thank the reviewer for this suggestion, in accordance with your comment, we added details of the abbreviation in the text as follows : “(Dénomination Orale 40 items : DO40)”.

  1. Add references to the outcome definition (lines 132-138).

R4 : thank you for noticing the missing references, we added them in the text.

  1. Review the citation of the statistical test (lines 161-163) in accordance with journal norms.

R5 : we thank the reviewer for highlighting this miss, we corrected it.

  1. It is not necessary to repeat results in the table and text. Remove the values in the text (172-175).

R6 : thank you for your comment, we removed this part in accordance with your suggestion as follows “We included 79 consecutive patients whose characteristics are detailed in the table 2 below.”

  1. Tables 3 and 5: it is missing two numbers after the commas.
  2. The results of MoCA could be described in Table 2 and the post-hoc description could be in the text. Table 4 is not necessary.

R8 : We would like to thank the reviewer for his advice, which enabled us to make substantial improvements to our article. The data in Table 4 relate to cluster analyses that appear after the description of the population. In this respect, we do not feel it is appropriate to include information on analyses that will take place afterwards in Table 2. We hope that the reviewer will agree with our position, but we are prepared to make any changes they deem necessary at a later stage.

  1. The aims of the study should be reviewed (lines 78-84). One of the aims was to compare the cognitive performance of three clusters and the second aim was to compare variables (neurological and demographic) that differentiate the groups. 

R9 : thank you for your comment, we modified the aims of our study as follows in the text : “The first objective of this study was to derive different cognitive profiles from complete neuropsychological data of adult patients with SCD. The second objective was to differentiate these patient groups based on demographics, medical history, and MoCA performance.”

  1. You cannot use the same MoCA cutoff for patients with lower and high schooling (lines 133-134). This can influence the results. Review the cutoff by schooling. 

R10 : thank you for your comment. We based our analysis with the adjusted MoCA score, according to schooling.

  1. The discussion is focusing on the results of the MoCA, but this is not informed previously in the aim, so this should be reviewed.

R11 : thank you for your remark, we corrected it in the main objectives of the study as follows in the manuscript : The third objective was to highlight the possibility of rapidly screening patients with cognitive disorders through a rapid comprehensive test such as the MoCA

Reviewer 2 Report

The approach of analysis is appreciable. However certain points need justification and elaboration.

In the introduction section, there is some good article related to neuro co agunation and sickle cell the author must cite these articles which makes them more robust and interesting to the reader.

1.       Maduakor, etal (2021). The Epidemiology of Neurological Complications in Adults With Sickle Cell Disease: A Retrospective Cohort Study. Frontiers in Neurology, 12. https://doi.org/10.3389/fneur.2021.744118

2.       lubusola Oluwole, et al Neurocognitive Assessment of Adults with Sickle Cell Disease: A Descriptive Study, Blood,Volume 138, Supplement 1,2021, https://doi.org/10.1182/blood-2021-145808.

3.       Sahu et al Neurocognitive Changes in Sickle Cell Disease: A Comprehensive Review. Annals of Neurosciences. 2022;0(0). doi:10.1177/09727531221108871

1.       Though the overall sample size is 79 however the PCA results are based on 66 samples only that is quite low sample size.

2.       It is suggested to give the Eigenvalues for each 5 factors and the sample variance contributed by each factors along with overall cumulative variance either in tabulated form or Scree plot.

3.       It is suggested to correlation matrix or the squared cosines table or correlation plots that will helps to understand the variables linked and the corresponding axis representing each factors. Also based on the result of PCA results of 66 individuals the hierarchical clustering of 79 individuals (including those not completed the neuro-cognitive tests) into three clusters need clarification and justification.

4.       There is no need of mentioning social cognition as it was not assessed

5.       Is the neuro-cognitive tests was all computer based or paper-pen tasks.

6.       What are the participant’s characteristics such as age, gender, sample size in oter group of participants that is “In order to analyze…….. …with SCD at the UMGGR”.

7.       Suggested to specify the Post hoc tests followed by Kruskall Wallis tests.

8.       What could be the probable reason for cluster 3 patients, is it the genotype exclusively or any other?

9.       The lines 279-281 (in discussion) should be checked (a patient with a total score of 26-20…….with a score above 20……cortical damages.

10.   Few spelling and typing mistakes should be corrected for uniformity (p<0.001/p < 0.001)

Author Response

The approach of analysis is appreciable. However certain points need justification and elaboration.

In the introduction section, there is some good article related to neuro co agunation and sickle cell the author must cite these articles which makes them more robust and interesting to the reader.

  1. Maduakor, etal (2021). The Epidemiology of Neurological Complications in Adults With Sickle Cell Disease: A Retrospective Cohort Study. Frontiers in Neurology, 12. https://doi.org/10.3389/fneur.2021.744118
  2. lubusola Oluwole, et al Neurocognitive Assessment of Adults with Sickle Cell Disease: A Descriptive Study, Blood,Volume 138, Supplement 1,2021, https://doi.org/10.1182/blood-2021-145808.
  3. Sahu et al Neurocognitive Changes in Sickle Cell Disease: A Comprehensive Review. Annals of Neurosciences. 2022;0(0). doi:10.1177/09727531221108871

Response : we are sorry for this missing reference. Indeed, at the time of our article writing we did not have this review and these two studies. We corrected this omission on the manuscript.

  1. Though the overall sample size is 79, however, the PCA results are based on 66 samples only that is quite low sample size.

R1: Recommendations on the appropriate sample size to use in a factor analysis are not lacking. In most cases, there is limited empirical evidence to support these recommendations. Mundfrom et al (2005) showed that sample size was stable when the variable-factor ratio was at least 6, but that the ratio could be much lower in cases of high communality. In our case, we have an overall communality of 13.68, i.e. an average communality of 0.72, which is very high. According to Mundfrom et al (2005), a sample size of 65 would be sufficient. According to Mundfrom et al (2005), a sample size of 65 would be sufficient. In addition, we mentioned in the methodology that we used a missing data replacement technique, using the MissMDA package of R software.

Mundfrom, D. J., Shaw, D. G., & Ke, T. L. (2005). Minimum Sample Size Recommendations for Conducting Factor Analyses. International Journal of Testing, 5(2), 159‑168. https://doi.org/10.1207/s15327574ijt0502_4

Following your commentary, we justify our analysis as follows in the text: “Of the 79 patients, 66 (84%), had completed all tasks of the neurocognitive assessment. The principal component analysis of neurocognitive test results with varimax rotation identified five distinct factors. The total commonality of the model is large (h²=13.68) indicating that the variance shared by all variables is substantially allowing for a factor analysis with a small sample size (Mundfrom et al., 2005).”

  1. It is suggested to give the eigenvalues for each 5 factors and the sample variance contributed by each factors along with overall cumulative variance either in tabulated form or Scree plot.

R2 : In order to improve the understanding of the factor analysis, and following the reviewer's advice, we propose to add as a supplement a table summarizing the initial eigenvalues and the scree plot and we notified it as follows in the manuscript :

“Of the 79 patients, 66 (84%), had completed all tasks of the neurocognitive assessment. The principal component analysis of neurocognitive test results with varimax rotation identified five distinct factors (see supplemental 1 for further information on the initial eigenvalues and scree plot).”

  1. It is suggested to the correlation matrix or the squared cosines table or correlation plots that will help to understand the variables linked and the corresponding axis representing each factor.

R3: The authors again thank the reviewer for his advice. The structure of the factors is described in Table 3 and the authors have taken care to name the factors according to the variables they contain. Thus, each factor name contains the cognitive functions to which the factor refers and the brain areas involved. We hope that these details are sufficient for the reader to have a clear idea of the meaning of each factor. We hope that the reviewer will agree with our position, but we would be prepared to make any changes the reviewer feels are necessary at a later stage.

  1. Also based on the result of PCA results of 66 individuals the hierarchical clustering of 79 individuals (including those not completed the neuro-cognitive tests) into three clusters need clarification and justification.

R4: We thank the reviewer for his precise remarks. Indeed, 66 patients completed all the tests while 13 patients completed only part of the tests. However, we employed a missing data replacement technique, using the MissMDA package included in the R software, as mentioned in our methodology. However, this missing data is pointed in the limitations of our study.

  1. There is no need of mentioning social cognition as it was not assessed.

R5 : thank you for your comment, we suppressed it in the table.

  • Is the neuro-cognitive tests was all computer based or paper-pen tasks.

R6 : thank you for your comment, you are right, it should be notified. We corrected it as follows in the text : “The cognitive battery combined standardized neuropsychological paper-pen tests that have an established utility in clinical practice.”

  1. What are the participant’s characteristics such as age, gender, sample size in oter group of participants that is “In order to analyze…….. …with SCD at the UMGGR”.

R7 : thank you for your comment.  the characteristics of these patients were not directly compared. However, patients were approached consecutively and offered participation  at the UMGGR clinic in order to analyze a diverse group of patients with and without cerebrovascular disease. Participants were adults of any SCD genotype who were able to provide informed consent and follow study instructions.

8Suggested to specify the Post hoc tests followed by Kruskall Wallis tests.

R8 : thank you for your comment, we added this precision in the text as follows : “post hoc Dwass, Steel, Critchlow and Fligner peer to peer comparisons”.

  1. What could be the probable reason for cluster 3 patients, is it the genotype exclusively or any other?

R9 : the authors thank the reviewer for his remark. Although there was no difference in the prevalence of the HbSS genotype compared to the other clusters, cluster 3 had more patients from France with a higher education level. This raises the question of access to education, which may be easier in France with possible accommodations both for the time-course of studies and for the taking of final exams. In accordance with his comment, we added it in the limitations of the study as follows : “Cluster 3 depicted as cognitively spared has a higher level of education, but it should be noted that the majority of patients in this cluster are from France. However, access to education and accommodation is often easier in France than in other countries, which may explain the higher level of education. But this also goes in the direction of our suggestions, namely that the disease disrupts the course of studies and that this is necessarily worser in countries which do not have the possibility of setting up specific arrangements in addition to appropriate rehabilitations.”

10.The lines 279-281 (in discussion) should be checked (a patient with a total score of 26-20…….with a score above 20……cortical damages.

R10 : thank you for your remark, we corrected it in the manuscript as follows : “Indeed, a patient with a total adjusted score greater than or equal to 26 could be considered cognitively unimpaired, a patient with a total adjusted score between 26 and 20 could be classified as mildly impaired, and a patient with a score less than 20 could be considered severely impaired with possible cortical damage.”

  • Few spelling and typing mistakes should be corrected for uniformity (p<0.001/p < 0.001)

R11 : thank you for your notice, we corrected it in the manuscript.